# Comprehensive GCMS and LC-MS/MS Metabolite Profiling of *Chlorella vulgaris*

**DOI:** 10.3390/md18070367

**Published:** 2020-07-17

**Authors:** Hamza Ahmed Pantami, Muhammad Safwan Ahamad Bustamam, Soo Yee Lee, Intan Safinar Ismail, Siti Munirah Mohd Faudzi, Masatoshi Nakakuni, Khozirah Shaari

**Affiliations:** 1Laboratory of Natural Products, Institute of Bioscience, Universiti Putra Malaysia, Serdang 43400, Selangor, Malaysia; hamza3983@gmail.com (H.A.P.); safwan.upm@gmail.com (M.S.A.B.); daphne.leesooyee@gmail.com (S.Y.L.); safinar@upm.edu.my (I.S.I.); sitimunirah@upm.edu.my (S.M.M.F.); 2Faculty of Science and Engineering, Soka University, 1-236 Tangi-machi, Hachioji City, Tokyo 192-8577, Japan; masatoshi-nakakuni@soka.gr.jp

**Keywords:** *Chlorella vulgaris*, biodiesel, phytoremediation, molecular networking, pigments

## Abstract

The commercial cultivation of microalgae began in the 1960s and *Chlorella* was one of the first target organisms. The species has long been considered a potential source of renewable energy, an alternative for phytoremediation, and more recently, as a growth and immune stimulant. However, *Chlorella vulgaris,* which is one of the most studied microalga, has never been comprehensively profiled chemically. In the present study, comprehensive profiling of the *Chlorella vulgaris* metabolome grown under normal culture conditions was carried out, employing tandem LC-MS/MS to profile the ethanolic extract and GC-MS for fatty acid analysis. The fatty acid profile of *C. vulgaris* was shown to be rich in omega-6, -7, -9, and -13 fatty acids, with omega-6 being the highest, representing more than sixty percent (>60%) of the total fatty acids. This is a clear indication that this species of *Chlorella* could serve as a good source of nutrition when incorporated in diets. The profile also showed that the main fatty acid composition was that of C_16_-C_18_ (>92%), suggesting that it might be a potential candidate for biodiesel production. LC-MS/MS analysis revealed carotenoid constituents comprising violaxanthin, neoxanthin, lutein, β-carotene, vulgaxanthin I, astaxanthin, and antheraxanthin, along with other pigments such as the chlorophylls. In addition to these, amino acids, vitamins, and simple sugars were also profiled, and through mass spectrometry-based molecular networking, 48 phospholipids were putatively identified.

## 1. Introduction

Microalgae are fast growing autotrophic organisms. They use light energy and carbon dioxide for the photosynthetic biomass production with higher efficiency compared to plants. There are over 300,000 species of microalgae, of which only around 30,000 (10%) of them have been documented [1]. They live in complex natural habitats, making them able to adapt rapidly in different extreme conditions such as variable salinity, temperature, nutrients and UV–irradiation. Therefore, they can produce a great variety of fascinating metabolites, with novel structures that possess biological activities which are not found in other organisms [2]. Microalgae produce some useful bio-products including β-carotene, astaxanthin, fucoxanthin, docosahexaenoic acid (DHA), eicosapentaenoic acid (EPA), bioactive and functional pigments, natural dyes, polysaccharides, amino acids, vitamins, antioxidants, and many more [3]. Algal research started in the 1950s, starting with studies on *Chlorella* and *Spirulina* species. *Chlorella* has been considered as an alternative in phytoremediation and potential source of renewable energy such as biodiesel due to its high lipid content [4]. Biodiesel production is primarily based on grease raw materials, such as palm oil, canola oil, animal tallow, and soybean oil [5]. However, the prices of these commodities are regulated by international laws which consequently affect biodiesel price, and about 80% of biodiesel costs vary with the price of grease raw materials [6], suggesting that new grease sources are needed in order to reduce the biodiesel price. As a result, there is growing interest in using microalgae for biodiesel production, especially because some produce about 150 times more oil per hectare than soybean [7]. Hence, the use of microalgae to produce biofuels has emerged as a promising alternative to the utilization of fossil fuels.

Single cell protein sources like microalgae have recently gained attention as protein sources for fish feed. In digestion trials with fish, microalgae are mostly fed in their natural form, grown without varying culture conditions or undergoing extraction [8]. Thus, microalgae are applied industrially in aquaculture hatcheries [9]. Normally fishes feed on microalgae as a natural source of nutrition. The microalgae can be safely used as a growth and immune stimulant because of their contents of essential amino acids, minerals, vitamins, chlorophylls, and some substances that have antioxidant capacities [10] like carotenoids and without worrying toxicity. Some microalgae are used mainly as healthy nutrition source in aquaculture in which they play important roles in the development of farming aquatic animals for healthy and profitable aquaculture practices [11,12].

Currently, most diseases of cultured fish are treatable with drugs [13]. However, the drugs used are usually expensive and are more effective in the initial therapy. Furthermore, their application is not entirely safe as they may result in several problems such as development of resistance, accumulation of hazardous residues in the fishes [14] and can also lead to water contamination [15]. In this regard, supplementing fish diet with microalgae as a way to enhance fish immunity and resistance to diseases has gained more attention since microalgae do not pose the problems associated with drugs usage in fish treatment [16]. Algal biomass also contains other valuable components, such as pigments, omega-3 fatty acids, minerals and vitamins, which further increase the nutritional value of algae-based feed [17]. Previous studies have shown that adding microalgae to animal feed resulted in enhancements in growth, physiological activity, feed utilization efficiency [18], stress response, starvation tolerance, disease resistance and carcass quality of the cultured fish [19]. These enhancements are very likely due to the diversity of metabolites present in the microalgae.

*Chlorella vulgaris* is a green-colored, single-celled, and spherical freshwater microalgae belonging to the family Chlorellaceae of the division Chlorophyta. It is highly valued as a food supplement for both humans and animals due to its high medicinal value derived from its carotenoid, chlorophyll and protein contents [20]. It was reported that the incorporation of *C. vulgaris* in fish diets increased prophenoloxidase activity, total hemocyte, and resistance against *Macrobrachium rosenbergii* post larvae and *Aeromonas hydrophila* infections [21]. In addition, increased growth performance and total fish protein were both observed by using fish feed supplemented with *Chlorella* [21]. Past research also reported that *C. vulgaris* is a potential source of renewable energy due to its high lipid content. However, the total lipid content is dependent on culture conditions [22]. In addition, studies using FTIR indicated that polyol and amide constituents in *C. vulgaris* played critical role in synthesizing palladium nanoparticles by reducing the metal ions in an eco-friendly and non-toxic process [23]. Moreover, the proven tolerance limits of *C. vulgaris* against various heavy metals and metalloids, detoxifying water supply contaminated with As^+3^ [24], are significant advantages of using *C. vulgaris* over other microalgal strains.

Previous studies on *C. vulgaris* have reported an abundance of metabolites, including lipids, fatty acids, amino acids, simple sugars, and few carotenoids. However, there is limited mass spectroscopic data on most of the metabolites of the microalgae species in general [25]. Various studies were conducted on *C. vulgaris* and other microalgae strains using GC-MS spectrometry and UV-Vis spectroscopy, while very few involved LC-MS spectrometry. However, all these studies were focused on a specific class of metabolites (usually carotenoids or lipids), obtained as a result of optimizing growth medium formulation or *via* mutagenesis, without comprehensively profiling the metabolites that occur as a result of normal media formulation. Usually, normal media formulations like the bold Basal’s media (BBM) (chemical composition) imitate the favorable natural conditions in which microalgae grow via metamorphosis. In addition, most of the researches also reported the unavoidable depletion of other metabolites in the course of enhancing the routes for the optimal production of the targeted metabolites of their research objective(s). For instance, *C. vulgaris* mutant strain was reported to have displayed superior lipid productivity with a minor PUFA level [26]. This is the same with total protein content of the wild species which solely depends on culturing conditions [27]. Under NO_3_ deficit, the production of photosynthetic pigments is drastically reduced with a huge decline of antioxidant activities particularly in *C. vulgaris* [28]. On the other hand, replete NO_3_ led to chlorophyll pigments and amino acid accumulation, while NO_3_ limitation produced a very low amino acid content but was very effective at increasing the generation of neutral lipid content [29,30]. Furthermore, most of the identification carried out on the carotenoids and chlorophylls was conducted on dietary supplements containing *C. vulgaris*. However, there is no information available regarding the comprehensive metabolite profile of *C. vulgaris* cultured under normal growing conditions and formulation which is ideal for incorporation into diets. It is important to note that the metabolite composition of a microalgae may contribute in one way or another to its biological effect [31]. Thus, having an appreciably good knowledge of the diversity in metabolite composition is a prerequisite to understanding their correlation to the nutritional and pharmacological effects of the microalgae. Therefore, it is equally important to comprehensively profile microalgae grown under normal conditions and formulations in order to gain a better insight into the *C. vulgaris* metabolome, which is the aim of the present study.

## 2. Results and Discussion

### 2.1. Identification of Fatty Acids in Chlorella vulgaris

Extracted fatty acids from *C. vulgaris* were derivatized to fatty acid methyl esters (FAMEs) *via* reaction with acetyl chloride [32] and subjected to GC-MS analysis (Figure 1A). From the *C. vulgaris*, nine saturated fatty acids ranging from C_14_ to C_22_ were detected, accounting for a percentage of 27.8% of the total fatty acids (Figure 1B), while the unsaturated fatty acids accounted for 71.2% of the total fatty acids. The compositional ratio of fatty acids was calculated from peak areas which were integrated in total ion chromatogram as shown in Figure 1.

The fatty acid profile of *C. vulgaris* is shown to be rich in omega-6, -7, -9, and -13 with omega-6 being the major fatty acid, representing more than sixty percent (>60%) of the total fatty acids (Figure 1D). This is a clear indication that this *Chlorella* species could serve as a good source of nutrition when incorporated in diets. The profile also showed that the fatty acid composition is mostly made up of C_16_-C_18_ (>92%) fatty acid (Figure 1C,D) which further supports *C. vulgaris* as a potential candidate for biodiesel production. These results are in agreement with the recently published work of Fernández-Linares et al. [33].

### 2.2. Metabolite Profiling of Chlorella vulgaris Ethanolic Extract

The ethanolic extract of *C. vulgaris* was analyzed by LC-MS/MS. The total scan PDA chromatogram and the total ion chromatogram of the extract are shown in Figure 2A,B, respectively. The observed peaks indicating the different metabolites present in the sample were labeled with numbers. A general idea of the compound classes contained in the extract was first obtained from the UV absorptions of compound peaks observed in the total scan PDA chromatograms (Figure 2A). Most of the UV-absorbing compounds detected by the photodiode array (PDA) detector showed maximum absorptions in the range of 200–300 nm. Nevertheless, carotenoids and chlorophyll pigments show characteristic absorption range of 200–400 nm and above, as a fingerprint that supports their identification. The identities of the metabolites were then further elucidated based on their molecular masses and mass fragmentation patterns (Appendix A). In some instances where the identity of a peak was masked by overlaps with other peaks of similar retention times, the extracted ion chromatogram (EIC) was obtained for metabolite identification, as in the case of astaxanthin (Appendix A). Although the LC-MS analysis was run in switching mode, almost all the compounds in the sample were better ionized in the positive mode. Therefore, identification of the compounds was conducted based on their full MS and MS/MS spectra obtained in positive ion mode.

#### 2.2.1. Identification of Carotenoids

The carotenoids were easily recognizable based on their typical **λ*_max_*** values of 410, 430, 440 and 460 nm [33,34,35,36,37,38,39], as shown in Table 2. The mass fragmentation pathway for carotenoid peaks was very useful in assigning the mass fragments and greatly aided their identification.

Peak 6 showed characteristics which were consistent with that of vulgaxanthin I, with a parent ion at *m/z* 340 [M + H]^+^ (observed = *m/z* 340.2592 and exact = *m/z* 340.1139, error *m/z* 0.1453). The peak occurred at t_R_ 3.98 giving MS/MS fragment ions at *m/z* 322 (a loss of 18 amu due to the elimination of water from the protonated parent ion), *m/z* 209 (product of α-cleavage of the pi-bond dissociated parent ion that undergoes electron loss), and *m/z* 84 (α-cleavage at the amide terminal of the molecule). A possible fragmentation pathway for vulgaxanthin I is as shown in Figure 3. The proposed fragment ions with *m/z* 339 and *m/z* 129, precursors to the fragment ions at *m/z* 209 and *m/z* 84, respectively, however were not detected by the MS detector. This could happen as a result of its rapid dissociation due to instability of the ions. Vulgaxanthin I has been previously identified in the higher plant *Berberis vulgaris* [40]. Although it has been reported before in brown algae *Chlorococcum humicola* [41], this is the first report of its occurrence in *C. vulgaris.*

Peaks 16 and 17 with t_R_ 14.35 and 14.52 were attributed to neoxanthin and violaxanthin, respectively. Both peaks showed similar parent ions at *m/z* 601.4 [M + H]^+^ and characteristic MS/MS fragment ions at *m/z* 583.4 [M + H − 18]^+^ due to loss of water molecule from the protonated parent ions. Both peaks also gave fragment ions at *m/z* 491.4 [M + H − 92 − 18]^+^, produced from a loss of C_7_H_8_ (92 amu: probably toluene) fragment along with one water molecule, and *m/z* 221.2 produced as a result of α-cleavage at one end of the molecules which occurred after a radical side rearrangement that precede the pi-bond dissociation of the parent ion (Figure 4). Two additional MS/MS fragments were identified with violaxanthin at very low intensities, where the fragment *m/z* 565.4 [M + H − 18 − 18]^+^ corresponding to the loss of two successive water molecules and *m/z* 509.4 [M + H − 92]^+^, which is unique to violaxanthin [41], corresponds to the loss of a lone C_7_H_8_ fragment. These two peaks share the same parent ion and some MS/MS product ions, the differences in some of their fragmentation pattern suggest that they are isomers. These fragmentation results agree with previously published results [42]. Fragmentation pathways proposed for neoxanthin and violaxanthin are provided in Appendix A, respectively.

Peak 18 with t_R_ 14.55–14.60 was attributed to antheraxanthin. This peak showed a parent ion at *m/z* 585 [M + H]^+^ and characteristic MS/MS fragments at *m/z* 567.4 [M + H − 18]^+^ due to loss of 18 amu corresponding to loss of water molecule, 475.4 [M + H − 92 − 18]^+^ corresponding to loss of C_7_H_8_ fragment along with a water molecule. The fragment *m/z* 221.1, was also produced with 100% intensity as expected [42]. Peak 20 at t_R_ 15.35–15.36, having a parent ion *m/z* 569.4 [M + H]^+^, exhibited two fragment ions with *m/z* 551.4 [M + H − 18]^+^ which occurred as a result of loss of one water molecule, and *m/z* 533.4 [M + H − 18 − 18]^+^ due to loss of two water molecules. This is characteristic of lutein and the absence of the fragment *m/z* 495, which would have resulted in further loss of 56 amu from the first fragment, suggested that the lutein identified in this sample is the isomer 15-*cis*-lutein [33]. Lutein fragmentation pathway was presented in Appendix A. Peak 29 at t_R_ 23.15–23.16 with a parent ion *m/z* 536.4 [M + H]^+^ and MS/MS fragment ions at *m/z* 444.4 [M + H − 92]^+^ and *m/z* 321.3 was attributed to β-carotene [42]. The fragmentation pathway for β-carotene is provided in Appendix A.

Peak number 26, which was not clearly seen in Figure 2B due to overlap, was identified from its EIC (Appendix A**)** which showed a peak at t_R_ 19.21 with parent ion at *m/z* 596.6 [M + H]^+^. The parent ion further dissociated to give fragment ions at *m/z* 578.6 [M + H − 18]^+^ and 560.6 [M + H − 18 − 18]^+^ corresponding to losses of 18 amu and 36 amu for successive loss of one and two water molecules, respectively. Based on this information, peak 26 was identified as astaxanthin [33]. The fragmentation pathway for astaxanthin is shown in Appendix A. The carotenoids detected in this work agree with recently published data [43,44].

#### 2.2.2. Identification of Chlorophyll Pigments

The chlorophylls are macromolecules with highly conjugated systems that do not always give MS/MS fragments data under the range of collision energy usually set for other molecules of lower masses. Hence their identification is best conducted with the help of their UV spectrum. Chlorophyll pigments (peaks 19 and 27–31) showed characteristic λ***_max_*** values in the range between 210 and 530 nm [35], as tabulated in Table 3, which also lists their respective parent ion masses. Compound identification made use of the mass fragmentation data provided in Appendix A (Appendix A).

Peak 19, the most intense peak in the UV PDA, at t_R_ 14.68 with characteristic *λ_max_* absorptions of 268, 474 and 536 nm was identified as pheophorbide-a based on the parent ion at *m/z* 593 [M + H]^+^ in which correspond to its MS base peak at t_R_ 14.74. Peak 27 is pheophytin-b with parent ion at *m/z* 885 [M + H]^+^ at MS t_R_ 21.13 (peak **27**) gives a UV peak at t_R_ 21.08 with absorption band of *λ_max_* 222, 436, 528 nm. Pheophorbide-b with parent ion *m/z* [M + H]^+^ 607 was assigned to peak 28. This peak was found under the MS t_R_ 21.25 which corresponds to the UV PDA observed peak at t_R_ 20.90 that show an absorption band of *λ_max_* 222, 436 and 528 nm. Pheophytin-a having an MS parent ion *m/z* 871 [M + H]^+^ at t_R_ 24.30 (peak 30**)** also shows a UV peak at t_R_ 24.28 with absorption band of *λ_max_* 408, 536 nm, corresponds with previously reported data [34]. The peak with t_R_ 28.70 (peak 31**)** gives a mass peak of *m/z* 894 [M + H]^+^corresponding to a UV peak at t_R_ 28.66 having an absorption of *λ_max_* 410, 538 nm, is attributed to chlorophyll a. The detected chlorophylic pigments agree with recently published data [44,45,46].

#### 2.2.3. Identification of Amino Acids, Fatty Acids, Lipids and Fatty Acyls

Table 4 shows peaks 2, 3, 5, 7 and 9 for the amino acids present in *C. vulgaris*. At t_R_ 1.44 leucine (peak 2) was observed to fragment into *m/z* 115, 86, 72 and 57. The characteristic MS/MS fragmentation pattern for peak 3 at t_R_ 2.29 with fragment ions *m/z* 165, 120, 103, 93, 91, 79, was attributable to phenylalanine. Peak 5 at t_R_ 2.72 with *m/z* 205 and 100% relative abundance was identified as tryptophan based on the fragment ions *m/z* 146, 144, 143, 142, 132, 118, 91, and 74. Lysophosphatidylethanolamine (Lyso-PE) was assigned to peak 7 at t_R_ 5.44 with fragment ion ions *m/z* 548, 452, 322, 209, 157, 114, 97, and 57. Disopyramide was the peak at t_R_ 6.30 with fragment ions *m/z* 322, 306, 212, 196, 114, and 74. Table 4 also shows the identified fatty acids, fatty acyls and lipids.

The identified amino acids, fatty acids and some of the lipids via LCMS agree with recently published results [47,48].

#### 2.2.4. Identification of Vitamins

Vitamin B-3 (nicotinic acid) was identified by fragment ions *m/z* 123, 80, 78, 53, and 45 which occurred due to peak 1 at t_R_ 1.33. Peak 4 at t_R_ 2.54 with characteristic fragmentation pattern *m/z* 142, 116, 103, 90, 87, 86, 73, 72, 70, 57, and 55 is attributed to vitamin B-5 (pantothenic acid).

#### 2.2.5. Identification of Other Compounds

The identification of simple sugars in the complex crude ethanol fraction of *C. vulgaris* was not very successful and hence very few were identified (Table 6). This may be attributed to many factors including the probability of many adduct formations which were not completely identified due to the large error values calculated in relation to the literature. (*R*)-cryptone, input *m/z* 139.1228 and exact *m/z* 139.1117 therefore error *m/z* 0.0111, is known to be found in the higher plant *Eucalyptus bosistoana* [49] and has never been reported before as a component metabolite of *C. vulgaris* in the literature. Hence further research is ongoing to establish more facts regarding its identification in *C. vulgaris.* The proposed mass fragment ions resulting from fragmentation of *R*-cryptone are shown in Figure 5. The detected simple sugars agree with previously published results [46].

#### 2.2.6. Identification of Lipids via Molecular Networking

Molecular networking (MN) assists in data mining *via* clustering of the MS/MS spectra based on fragmentation cosine similarities [50,51]. The molecular network of the ethanol extract was generated in order to analyze the lipid content of *C. vulgaris* more comprehensively, to enrich the information obtained from the fatty acid analysis *via* GC-MS. Therefore, the typical nature of the lipid content can be fully viewed *via* a putative annotation of the different lipids that make up the important property of *C. vulgaris* as a potential candidate for nutrition and biodiesel production using MN. Figure 6 shows the generated MN with the different clusters, each cluster shared some distinct fragments and fragmentation pattern.

The putative annotation was conducted with reference to different mass spectroscopic data bases. Three major and distinct clusters were observed in the network comprising diacylglycerophosphoserines, diacylglycerophosphocholines, and glycosphingolipids clusters. Another cluster comprising several separate sub-clusters of monoacylglycerophosphoethanolamines was also detected. Identification of the metabolites was performed based on the systematic study of the fragmentation pathways and patterns observed from the resulting network. The mass spectrometry-based MN allowed the identification and putative annotation of 48 lipids in the major clusters as shown in Table 7. The MN was capable not only to dereplicate known lipids, but also pointed out related derivatives, described for the first time in *C. vulgaris*. The resulting annotation from MN conforms with those obtained from GCMS fatty acid analysis in terms of the number and degree of unsaturation; most of the lipids identified from the MN contain the substituent fatty acid carbon ring ranging from C_16_-C_18_ which mostly belong to omega 6 fatty acids containing 2–3 degrees of alkene unsaturation.

Visual inspection of the *C. vulgaris* MN showed that glycosphingolipids clustered in agreement with their substituent’s similarities, i.e., grouped according to the type and size of the attached fatty acids. All the glycosphingolipids identified are mainly disubstituted lipids and mostly contain dialkene with few saturated, mono- and tri-alkene fatty acid substituents. Some phosphoethanolamines which were totally absent in their class cluster, were also seen in this cluster sharing some common fragments with glycosphingolipids. Therefore, the typical nature of the lipid content can be fully viewed via a putative annotation of the different lipids that make up the important property of *C. vulgaris* as a potential candidate for nutrition and biodiesel production using MN.

Some of the identified lipids agree with previously published results [52,53].

The fragmentation pattern in phosphoserine cluster (Figure 7) is observed by the π-bond dissociation of one of the alkene bonds of the corresponding fatty acid to yield a free radical ion that eventually leads to the elimination of C_2_H_5_ or C_3_H_7_ via б-bond dissociation. These fragments correspond to a loss of 29 and 43 amu, respectively. The fragment *m/z* 837 was a result of radical-site rearrangement that led to the elimination of two subsequent H_2_O molecules via α-bond cleavage corresponding to a loss of 18 amu. In addition, fragment *m/z* 593 mostly occurred as a result of the complete elimination of one of the fatty acid substituents as a result of γ-charge-site rearrangement. Fragments *m/z* 178 and *m/z* 113 were due to б-bond dissociation eliminating two alkene bonds involving 13 C-atoms and or without the alkene bonds involving only 8 C-atoms respectively as seen in Figure 8.

The fragmentation pattern in diacylglycerophosphocholines (Figure 9) is slightly different from the other classes because most of the commonly shared fragments among the lipids of this group contain the phospho-terminal of the lipid molecule. For instance, fragment *m/z* 189, which was a result of charge-remote rearrangement, contained both the phospho-terminal as well as two hydroxyl groups (OH). Other fragments *m/z* 137, 125, 99, and 81 contain only the phospho-terminal in different ionic forms which are also products of charge-remote rearrangements, and in few instances, from inductive cleavages, as shown in Figure 10.

The largest cluster of the MN, glycosphingolipids (Figure 11), are mainly characterized by the presence of one or two tetrahydropyran rings within the fragments and the presence of mono-alkene substituted fatty acids, as shown in Figure 12.

## 3. Materials and Methods

### 3.1. Fresh Water Microalgae Culture

*Chlorella vulgaris* was obtained from the Aquatic Laboratory Faculty of Veterinary Medicine, Universiti Putra Malaysia and prepared by raising 200 L of cultured media from 40 milliliters of starter-culture using Bold Basal’s media (BBM). The whole process was conducted in duplicate (100 L each). The upscaling of culture was done *via* the addition of ten liters of culturing media in three days interval after the growth is checked using hemacytometer under a 400× microscope. The parameter for the algal culture was adopted from the original BBM media with slight modification in temperature which we could not maintain at 28 °C (changes in the range of 25–30 °C). The microalgae biomass production was cultivated in fresh water containing the prepared BBM media having various concentrations of different acids and trace elements autoclaved solutions. The growth medium was adjusted and monitored at pH 7.5. The wet crude product was harvested via centrifugation by employing high-speed Sorvall Evolution RC centrifuge (Thermo Electron Corporation, Asheville, NC, USA) at 12,000 rpm at 25 °C, kept at −80°C for five days, and lyophilized using a freeze drier [48]. Solvent extraction was carried out with ethanol and for LC-MS/MS analysis.

### 3.2. Solvents and Chemicals

Analytical grade methanol, ethanol, ethyl acetate, chloroform and hexane were purchased from Merck Millipore (Darmstadt, Germany).

### 3.3. Microalgae Solvent Extraction Procedure

Lyophilized microalgae biomass (100 mg) was dissolved in 50 mL ethanol and vortexed for 5 min. The solution was then extracted *via* sonication for 30 min using ultrasonic water bath (SK8210HP Shanghai KUDOS Ultrasonic Instrument Co. Ltd., Shanghai, China) at room temperature. The solvent extract was then filtered through Whatman No. 1 filter paper and the procedure is repeated with another 50 mL of ethanol for a second and third round of extraction. The filtered extracts were pooled and evaporated to dryness using a rotary evaporator (Heidolph Instruments GmbH amd Co.KG, Schwabach, Germany) at 30 °C, and stored at −20 °C until further analysis. The ethanol fraction was then used for LC-MS analysis.

### 3.4. Extraction of Fatty Acids for GC-MS Analysis

Fatty acids in the crude sample of *C. vulgaris* (~3 mg) were extracted using chloroform and methanol (1:2 v/v) by an ultrasonic device for 10 min. To separate the residue, the mixture was separated by centrifugation at 3000 rpm for 10 min, and the liquid phase was transferred into a glass tube. These extraction steps were repeated three times to obtain maximum extract. After the extraction, several ml of Milli Q water was added, the glass tube was shaken using vortex. Then, to separate impurities (which were dissolved in the water) and the organic solvent phase containing fatty acids, the glass tube was centrifuged at 3000 rpm for 10 min. The water was removed, and the organic solvent phase was dried once. Derivatization into FAMEs were performed with acetyl chloride and methanol (5:100 v/v) [32]. A 10ml acetyl chloride and methanol was added into the dried glass tube, and the derivatization was achieved by reacting under heat at 100 °C for 60 min using a heat block. After cooling at room temperate, the glass tube was added 2 ml of hexane and shake well, and then an aliquot of the upper phase was transferred to a new glass vial. This extraction step was repeated two more times. The final removal of hexane was performed on a 40 °C hot plate with nitrogen stream. Then, the sample was resolved with an accurate amount of hexane (300 µL). Finally, 1 μL of the sample was injected into a GC–MS for fatty acid analysis. Using this method, non-esterified fatty acids were obtained [49]. The identification of FAMEs was assisted by National Institute of Standards and Technology Library (NIST 17 version 2.3) and was confirmed manually. The mono-unsaturated fatty acids were manually detected by the presence of intense peaks of *m/z* 55 and *m/z* 69 along with the molecular weight peak. Meanwhile, the di-unsaturated fatty acids were detected by the presence of *m/z* 55 and *m/z* 67 intense peaks and *m/z* 74 and *m/z* 87 intense peaks were features of saturated fatty acids.

### 3.5. Sample Preparation for UHPLC-MS/MS Analysis

The ethanol extract (2 mg) was dissolved in LCMS-grade methanol (1 mL). Dissolved extract was vortexed for 10 min, centrifuged for 10 min and filtered through a nylon filter (0.22 µm) into a glass vial for LC-MS/MS analysis [50].

### 3.6. GC-MS Analysis

The extracted fatty acids were analyzed using a gas chromatography (GC, HP 6890)-mass spectrometry (MS, HP 5973) (Hewlett Packard, Palo Alto, CA, USA). The GC column was set ZB-WAX column, 30 m (length) × 0.25 mm (I.D) × 0.25 μm film thickness. Helium was used as the carrier gas with a flow rate of 1.0 mL/min. The injector was set at 220 °C in split mode. The temperature gradient of the GC oven started with a 70 °C initial temperature, a linear increase to 170 °C at 11 °C /min, a slower linear increase to 175 °C at 0.8 °C /min, followed by an increase to reach 220 °C at 20 °C/min and a final 2.5 min hold. The total run time was 60 min. The MS quadrupole and MS ion source were programmed at 150 °C and 230 °C, respectively. MS data were obtained with scan mode scanning from 20 to 450 amu.

Analysis of MS data for the fatty acids was conducted using enhanced MSD ChemStation E.02.02.143 (Agilent Technologies, Inc. Santa Clara, CA, USA) fitted with NIST Library database (NIST 17 Version 2.3). The compositional rate in fatty acids was calculated from a comparison of each area in the fatty acids taken from the total ion chromatogram.

### 3.7. UHPLC-MS/MS Analysis

The crude methanol extract was separated using a C18 Reversed-phase Hypersil GOLD aQ column (100 × 2.1 mm ~1.9 µm) (Thermo, Waltham, MA, USA) at 30 °C on Dionex Ultimate 3000 UHPLC with a diode-array DAD-3000 detector (Thermo Fisher Scientific, Waltham, MA, USA). Gradient elution was performed with LC-MS grade Solvent A (0.1% formic acid and 10 mMol ammonium formate in 500 mL methanol (70%) and acetonitrile (30%)) and Solvent B (0.1% formic acid and 10 mMol ammonium formate in 500 mL water) for the following gradient: 20% A in 5 min, and 20–80% A in the next 25 min at a flow rate of 0.2 mL/min. The concentration of sample extract was 1 mg/mL and the injection volume were set to 10 µL and the UV detector was set at 210, 310, 410 and 510 nm. The MS analysis was done on Q-Exactive Focus Orbitrap LC-MS/MS system. The eluent was monitored by ESI-MS under positive and negative switching mode and scanned from *m/z* 100 to 1500 amu. ESI was conducted using a spray voltage of 4.2 kV. High purity nitrogen gas was used as dry gas at a sheath gas flow rate of 40 (arbitrary units) and aux gas flow rate of 10 (arbitrary units). Capillary temperature was set at 350 °C while aux gas heater temperature was set at 10 °C. The MS data analysis was conducted using ThermoXcalibur 2.2 SP1.48 (Thermo Fisher Inc. Waltham, MA, USA) and literature data. Further, the confirmation of newly identified compounds was supported by its mass error that shows relatively low values indicating the high possibility of correct identification of the newly reported compounds.

The mass spectrometry molecular networks were created using the Global Natural Products Social Molecular Networking (GNPS) platform (http://gnps.ucsd.edu) [51]. The MS data were first converted into mzXML format using MSConvert [51]. Spectral information generated was uploaded on GNPS using FileZilla and was used to generate an MS/MS molecular network using the GNPS Data Analysis Workflow. The precursor ion mass tolerance was set to 0.02 Da and a fragment ion mass tolerance of 0.02 Da. The fragment ions below 10 counts were removed from the MS/MS spectra. The MN were generated using 6 minimum matched peaks and a cosine score of 0.7. The resulting data were downloaded and visualized using Cytoscape 3.7.1 software (Institute of Systems Biology Seattle, Washington D.C., USA) [50].

## 4. Conclusions

MS-based metabolite profiling of crude sample of *C. vulgaris* microalgae grown in normal conditions, proved to be efficient in the determination of certain metabolites including carotenoids, amino acids, vitamins and other pigments such as the chlorophylls. The combination of methanol 70% and acetonitrile 30% as the organic solvent used proved to be efficient in determining these compounds. In total, 31 metabolites were successfully determined using LC-MS among which two of them, vulgaxanthin I and R-crypton, were never reported in *C. vulgaris* before. In addition, 48 lipids were putatively identified via MN. On the other hand, the fatty acid content was best determined using GC-MS via fatty acid analysis due to their volatility and ease of derivatization into FAME, and a total of 20 fatty acids were identified in the derivatized sample with higher percentage of unsaturation and omega-6 being the most dominant. Recent novel bioinformatics approaches such as the MN and in-silico fragmentation tools have emerged and provided a new perspective for early metabolite identification in natural products research. Thus, in this research, an efficient exploitation of datasets was employed for automated data treatment and access to dedicated fragmentation databases during MN. As a result, a larger profile of *C. vulgaris* was successfully established and many lipids were putatively identified which were not reported before.

The presence of carotenoids, chlorophylls and amino acids suggests that the candidate sample *C. vulgaris* is a good source of nutrition supported by the presence of omega-6, -7, -9, and -13 fatty acids. Also, the fatty acid profile suggests that *C. vulgaris* is a good candidate for biodiesel production. More studies need to be carried out in order to determine the correct proportion of nutrient composition in this promising microalga.

## Figures and Tables

**Figure 1 marinedrugs-18-00367-f001:**
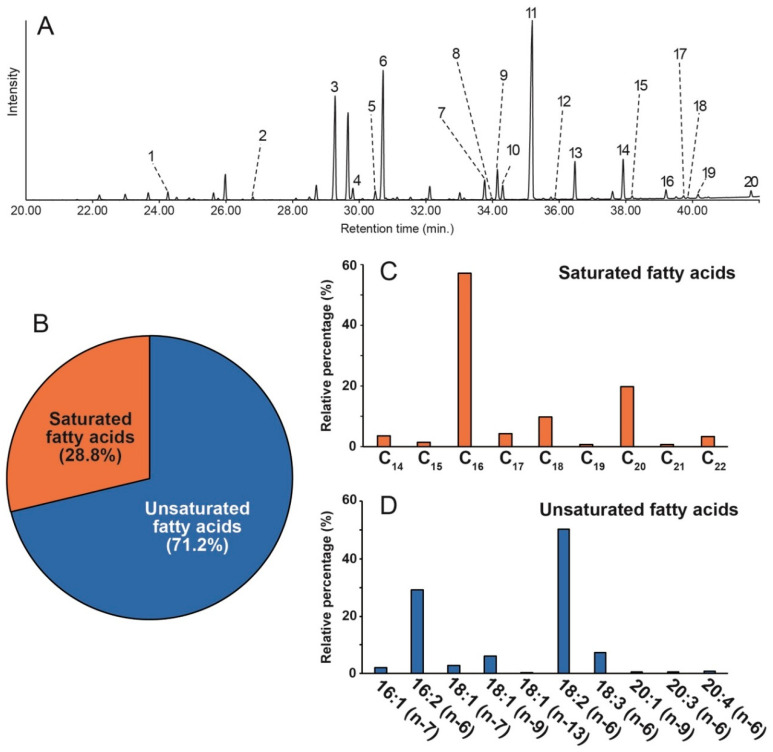
(**A**) Total ion chromatogram of extracted fatty acid methyl esters in *Chlorella vulgaris* obtained by GC−MS analysis (numbers correspond to peak numbers in Table 1) (**B**) The percentage composition of saturated and unsaturated fatty acid methyl esters (**C**) Percentage distribution of individual saturated fatty acids relative to the total saturated fatty acid content (**D**) Percentage distribution of individual unsaturated fatty acids relative to the total unsaturated fatty acid content.

**Figure 2 marinedrugs-18-00367-f002:**
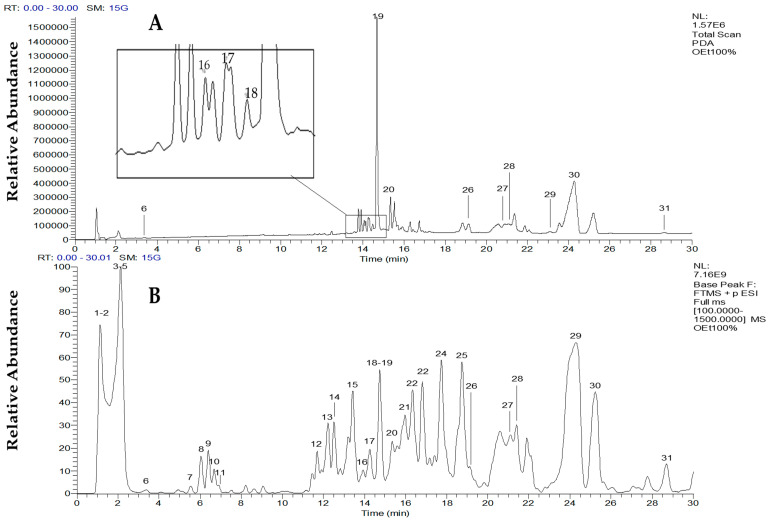
(**A**) Total scan PDA chromatogram and (**B**) total ion chromatogram (TIC) in positive ion mode of *Chlorella vulgaris*. The number above each peak represent peak numbers, corresponding to the peak numbers in Table 2, Table 3, Table 4 and Table 5.

**Figure 3 marinedrugs-18-00367-f003:**
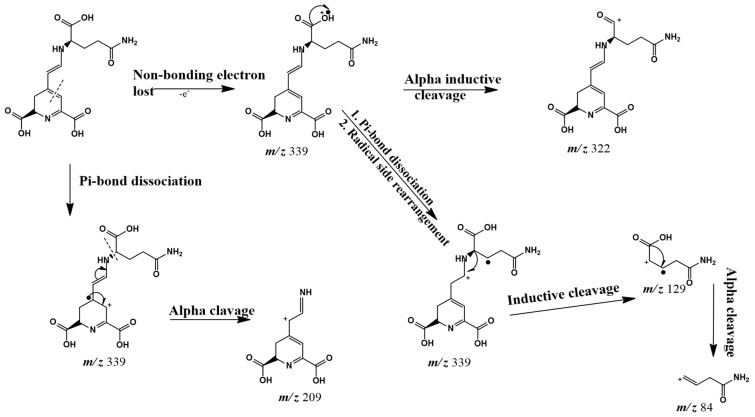
Proposed fragmentation pathway for vulgaxanthin I.

**Figure 4 marinedrugs-18-00367-f004:**
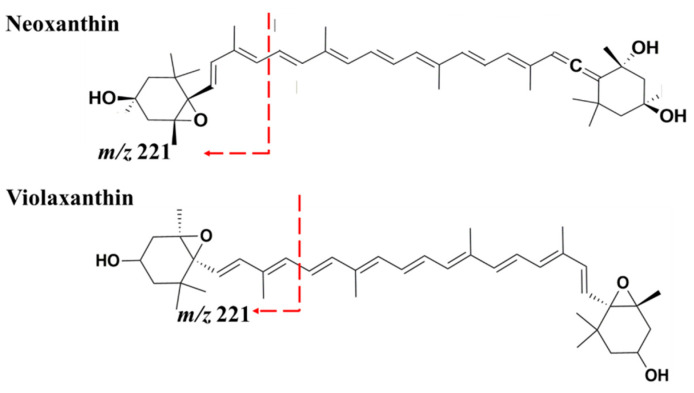
Proposed fragmentation pathways for neoxanthin and violaxanthin, producing *m/z* 221 fragment ions.

**Figure 5 marinedrugs-18-00367-f005:**
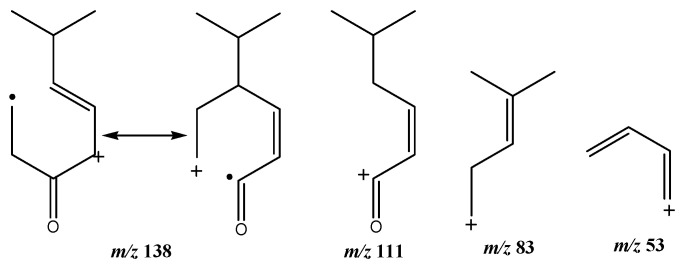
Proposed mass fragments resulting from fragmentation of *R*-cryptone**.**

**Figure 6 marinedrugs-18-00367-f006:**
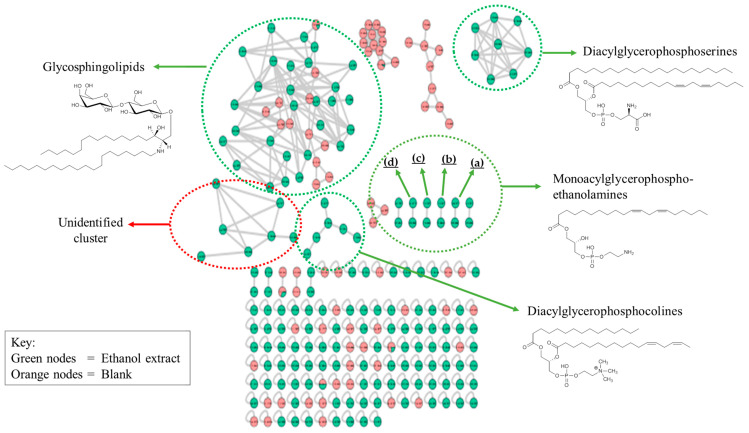
Full molecular network of *Chlorella vulgaris* ethanol extract showing lipid clusters consisting of diacylglycerophosphoserines, diacylglycerophosphocholines, glycosphingolipids, several small clusters of monoacylglycerophosphoethanolamines (a,b,c,d) and an unidentified cluster. Structures shown are representative examples of the lipids in each cluster.

**Figure 7 marinedrugs-18-00367-f007:**
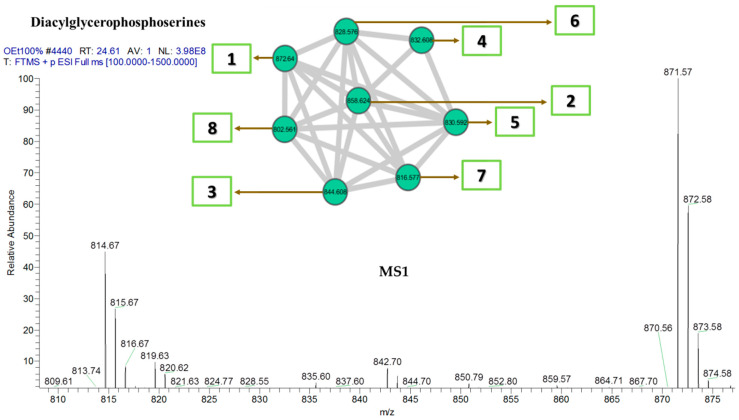
Diacylglycerophosphoserine cluster with MS1 showing the nodes. Numbers correspond to lipid number as listed in Table 7.

**Figure 8 marinedrugs-18-00367-f008:**
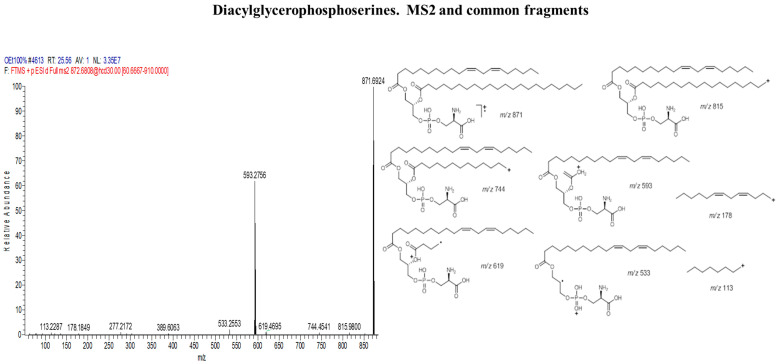
Diacylglycerophosphoserines cluster(1); some of the common fragments within the cluster.

**Figure 9 marinedrugs-18-00367-f009:**
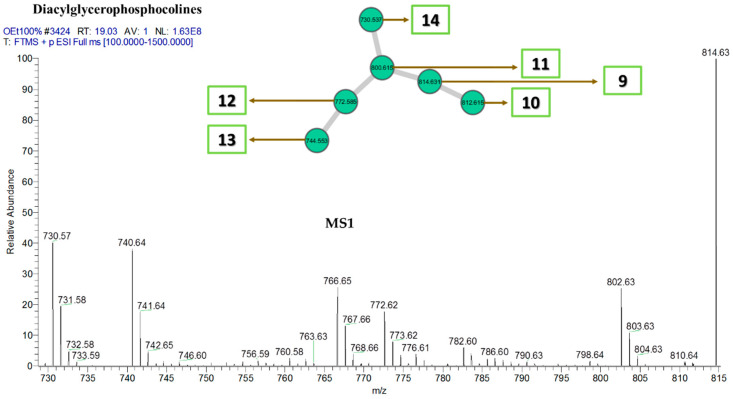
Diacylglycerophosphocholine cluster with MS1 showing the nodes. Numbers correspond to lipid number as listed in Table 7.

**Figure 10 marinedrugs-18-00367-f010:**
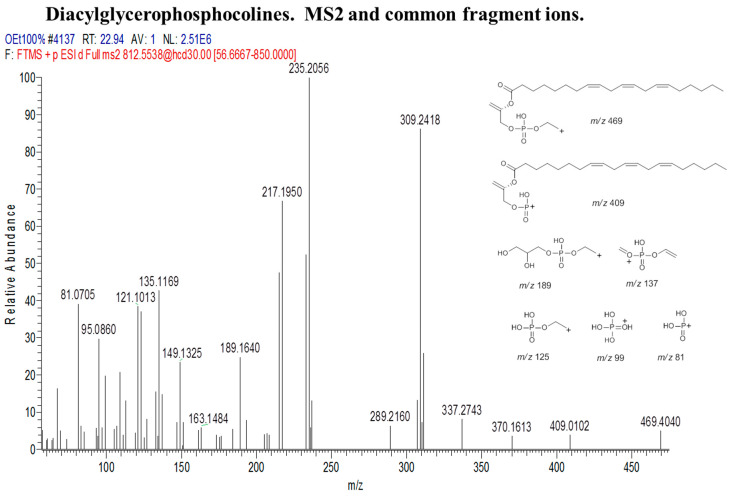
Some common fragments within the diacylglycerophosphocholine cluster based on MS2.

**Figure 11 marinedrugs-18-00367-f011:**
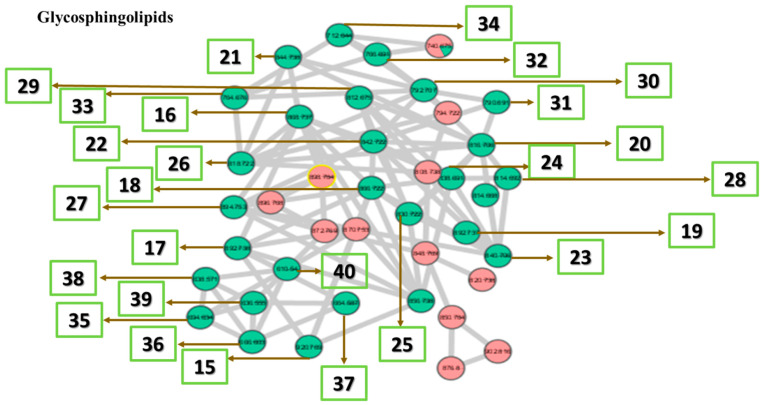
Glycosphingolipid cluster with MS1 showing the nodes. Numbers correspond to lipid number as listed in Table 7.

**Figure 12 marinedrugs-18-00367-f012:**
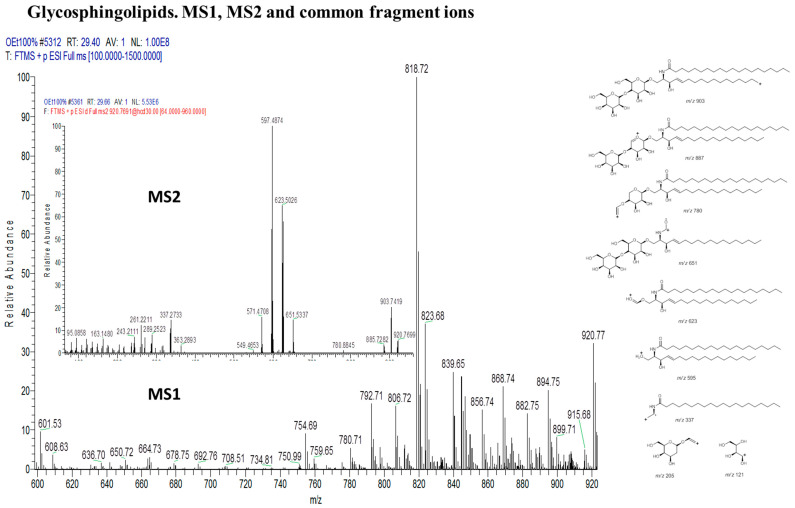
Some common fragments within the glycosphingolipid cluster based on MS1 and MS2.

**Table 1 marinedrugs-18-00367-t001:** Fatty acid composition of *Chlorella vulgaris.* The table shows the peak numbers, the corresponding systematic and trivial names of the fatty acid, designation and percentage composition.

Peak	Systematic Name	Trivial Name	Designation	Composition (%)
1	Tetradecanoic acid	Myristic acid	C_14_	1.0
2	Pentadecanoic acid	-	C_15_	0.4
3	Hexadecanoic acid	Palmitic acid	C_16_	16.4
4	9-Hexadecenoic acid	Palmitoleic acid	C_16:1 [n-7]_	1.5
5	Heptadecanoic acid	Margaric acid	C_17_	1.2
6	7,10-Hexadecadienoic acid	-	C_16:2 [n-6]_	20.4
7	Octadecanoic acid	Stearic acid	C_18_	2.8
8	5-Octadecenoic acid	-	C_18:1 [n-13]_	0.3
9	9-Octadecenoic acid	Oleic acid	C_18:1 [n-9]_	4.2
10	11-Octadecenoic acid	-	C_18:1 [n-7]_	2.0
11	9,12-Octadecadienoic acid	Linoleic acid	C_18:2 [n-6]_	35.1
12	Nonadecanoic acid	Nonadecylic acid	C_19_	0.2
13	9,12,15-Octadecatrienoic acid	Linolenic acid	C_18:3 [n-6]_	5.2
14	Eicosanoic acid	Arachidic acid	C_20_	5.7
15	9-Eicosenoic acid	-	C_20:1 [n-9]_	0.4
16	Eicosadienoic acid	-	C_20:2_	1.3
17	11,14,17-Eicosatrienoic acid	Homolinolenic acid	C_20:3 [n-6]_	0.4
18	Heneicosanoic acid	Heneicosylic acid	C_21_	0.2
19	5,8,11,14-Eicosatetraenoic acid	Arachidonic acid	C_20:4 [n-6]_	0.6
20	Docosanoic acid	Behenic acid	C_22_	0.9

**Table 2 marinedrugs-18-00367-t002:** Carotenoids identified in *Chlorella vulgaris*. The underlined fragment ion is unique to the corresponding compound**.** The table shows peak number, name of the carotenoid, retention time in minutes (t_R_), characteristic absorption frequencies in nanometers (*λ_max_*), parent ion [M + H]^+^ mass per charge ratio, MS/MS mass per charge ratio of fragment ions, and references.

Peak	Carotenoid	t_R_ (min)	λ*_max_* (nm)	[M + H]^+^ (*m/z*)	MS/MS (*m/z*)	References
6	Vulgaxanthin I	3.98	424, 476	340	322 (M + H − 18)^+^ (20%) 209 (100%), 84 (3%)	Not previously reported in *C. vulgaris*
16	Neoxanthin	14.35	414, 436, 462	601.4	583.4 (M + H − 18)^+^ (8%) 491.4 (M + H − 92 − 18)^+^ (10%) 221.2 (100%)	[33,34,36]
17	Violaxanthin	14.52	416, 442	601.4	583.4 (M + H − 18)^+^ (6%) 565.4 (M + H − 18 − 18)^+^ (4%) 509.4 (M + H − 92)^+^ (4%) 491.4 (M + H − 92 − 18)^+^ (7%) 221.2 (100%)	[33,34,36,37,38]
18	Antheraxanthin	14.55–14.60	400, 424	585.4	567.4 (M + H − 18)^+^ (35%) 475.4 (M + H − 92 − 18)^+^ (10%) 221.2 (100%)	[35,38]
20	Lutein	15.35–15.36	414, 438, 464	569.4	551.4 (M + H − 18)^+^ (25%) 533.4 (M + H − 18 − 18)^+^ (5%)	[32,33,34,36]
26	Astaxanthin	19.21	372, 436	596.6	578.6 (M + H − 18)^+^ (100%) 560.6 (M + H − 18 − 18)^+^ (10%) 284.3 (50%)	[39]
29	β-carotene	23.15–23.16	412, 440, 468	536.4	444.4 (M + H − 92)^+^ (20%) 321.3 (18%)	[33,34,39]

**Table 3 marinedrugs-18-00367-t003:** Chlorophylic pigments identified in *Chlorella vulgaris**.*** The table shows peak number, the corresponding pigment name, retention time in minutes (t_R_), characteristic absorption frequencies in nanometers (*λ_max_*), parent ion [M + H]^+^ mass per charge ratio, and references.

Peak	Pigment	t_R_ (min)	λ_max_ (nm)	[M + H]^+^ (*m/z*)	References
19	Pheophorbide-a	14.68	268, 474, 536	593	[34]
27	Pheophytin-b	21.08	222, 436, 528	885	[34]
28	Pheophorbide-b	21.37	202, 222, 372, 436, 528	607	[34]
30	Pheophytin-a	24.28	204, 408, 536	871	[34]
31	Chlorophyll-a	28.66	202, 410, 538	894	[34]

**Table 4 marinedrugs-18-00367-t004:** Amino acids, fatty acids, lipids and fatty acyls composition of *Chlorella vulgaris*. Underlined *m/z* values indicate intense fragments**.** The table shows peak numbers, corresponding compound name, retention time in minutes (t_R_), parent ion [M + H]^+^ mass per charge ratio and MS/MS mass per charge ratio of fragment ions.

Peak	Amino Acid	t_R_ (min)	[M + H]^+^ (*m/z*)	MS/MS (*m/z)*
2	Leucine	1.44	132	115, 86, 72, 57
3	Phenylalanine	2.29	166	165, 120, 103, 93, 91, 79
5	Tryptophan	2.72	205	159, 146, 144, 143, 142, 132, 118, 91, 74
7	Lysophosphatidylethanolamine [Lyso-PE]	5.44	566	548, 452, 322, 209, 157, 114, 97, 57
9	Disopyramide	6.30	340	322, 306, 212, 196, 114, 74
	**Fatty acids**			
13	Fatty acid	12.22	492	474, 309, 258, 184, 141, 138, 124, 112, 104, 102, 86, 78, 70, 60
21	Nonadecanoyloxyoctadeca- trienoic acid	15.98	575	447, 263, 239, 221, 161, 137, 109, 95, 81, 71, 69, 57, 55
12	1-Acetoxy-2-hydroxy-16- heptadecyn-4-one	11.70	325	233, 215, 175, 173, 145, 135, 121, 109, 95, 93, 91, 81, 67
14	Hexadecatrienal	12.51	235	226, 211, 173, 153, 133, 119, 111, 109 105, 95, 91, 81, 79, 67, 55 53
	**Lipids**			
15	1-[9*Z*,12*Z*-octadecadienoyl]-sn- glycero -3-phosphocholine	13.44	520	502, 484, 342, 337, 260, 184, 125, 104, 86, 60
22	1-hexadecanoyl-2-[5*Z*,8*Z*,11*Z*, 14*Z*-eicosatetraenoyl]-sn- glycero-3-phosphocholine	16.35	782	782, 184, 124, 184, 86
23	1-tetradecanoyl-2-[11*Z*,14*Z*- eicosadienoyl]-glycero-3- phosphocholine	16.80	758	553, 357, 199, 184, 124, 86
24	1-hexadecanoyl-2-[11*Z*,14*Z*- eicosadienoyl]-glycero-3- phosphocholine	17.74	786	688, 552, 501, 474, 215, 184, 125, 104, 87, 86, 60
25	1-octadecanoyl-2-[11*Z*,14*Z*- eicosadienoyl]-sn-glycero-3- phosphocholine	18.74	814	673, 614, 513, 462, 338, 184, 130, 104, 86

**Table 5 marinedrugs-18-00367-t005:** Composition of vitamins in *Chlorella vulgaris*. Underlined *m/z* values indicate intense fragments**.** The table show peak numbers, corresponding vitamin name, retention time in minutes (t_R_), parent ion [M + H]^+^ mass per charge ratio and MS/MS mass per charge ratio of fragment ions.

Peak	Vitamin	t_R_ (min)	[M + H]^+^ (*m/z*)	MS/MS (*m/z)*
1	Vitamin B-3 [nicotinic acid]	1.33	124	123, 80, 78, 53, 45
4	Vitamin B-5 [Pantothenic acid]	2.54	220	142, 116, 103, 90, 87, 86, 73, 72, 70, 57, 55

**Table 6 marinedrugs-18-00367-t006:** Identified simple sugars and *R*-cryptone in *Chlorella vulgaris**.*** Underlined *m/z* values indicate intense fragments**.** The table shows peak numbers, corresponding compound name, retention time in minutes (t_R_), parent ion [M + H]^+^mass per charge ratio, MS/MS mass per charge ratio of fragment ions.

Peak	Compound	t_R_ (min)	[M + H]^+^ (*m/z*)	MS/MS (*m/z)*
8	*(R)-*cryptone	5.69	139	138, 111, 83 53
10	Glucose	6.54	181	163, 145, 121, 119, 117, 115, 105, 83, 59, 57, 55
11	Galactose	7.19	181	163, 145, 135, 133, 121, 119, 117, 115, 105, 103, 91, 87, 73, 59, 57, 55

**Table 7 marinedrugs-18-00367-t007:** Putative annotation of the lipids from ethanol extract of *Chlorella vulgaris**.*** The table shows 48 lipids, their names under each lipid class, molecular formula (MF) and corresponding parent ion [M + H]^+^ mass per charge ratio.

No.	Lipids	MF	*m/z*
	**Diacylglycerophosphoserines**		
1	1-[11*Z*,14*Z*-eicosadienoyl]-2-docosanoyl-glycero-3-phosphoserine	C_48_H_90_NO_10_P	872
2	1-[11*Z*,14*Z*-eicosadienoyl]-2-heneicosanoyl-glycero-3-phosphoserine	C_47_H_88_NO_10_P	858
3	1-[11*Z*,14*Z*-eicosadienoyl]-2-eicosanoyl-glycero-3-phosphoserine	C_46_H_86_NO_10_P	844
4	1-[11*Z*-eicosenoyl]-2-nonadecanoyl-glycero-3-phosphoserine	C_45_H_86_NO_10_P	832
5	1-[11*Z*,14*Z*-eicosadienoyl]-2-nonadecanoyl-glycero-3-phosphoserine	C_45_H_84_NO_10_P	830
6	1-[11*Z*,14*Z*-eicosadienoyl]-2-[9*Z*-nonadecenoyl]-glycero-3-phosphoserine	C_45_H_82_NO_10_P	828
7	1-[11*Z*,14*Z*-eicosadienoyl]-2-octadecanoyl-glycero-3-phosphoserine	C_44_H_82_NO_10_P	816
8	1-[11*Z*,14*Z*-eicosadienoyl]-2-heptadecanoyl-glycero-3-phosphoserine	C_43_H_80_NO_10_P	802
	**Diacylglycerophosphocholines**		
9	1-octadecanoyl-2-[11*Z*,14*Z*-eicosadienoyl]-sn-glycero-3-phosphocholine	C_46_H_88_NO_8_P	814
10	1-octadecanoyl-2-(5Z,11Z,14Z-eicosatrienoyl)-sn-glycero-3-phosphocholine	C_46_H_86_NO_8_P	812
11	1-[9*Z*-octadecenoyl]-2-[9*Z*-nonadecenoyl]-glycero-3-phosphocholine	C_45_H_86_NO_8_P	800
12	1-octadecanoyl-2-[9*Z*,12*Z*-heptadecadienoyl]-glycero-3-phosphocholine	C_43_H_82_NO_8_P	772
13	1-[9*Z*-octadecenoyl]-2-[9*Z*-pentadecenoyl]-glycero-3-phosphocholine	C_41_H_78_NO_8_P	744
14	1-[9*Z*-octadecenoyl]-2-[9*Z*-tetradecenoyl]-glycero-3-phosphocholine	C_40_H_76_NO_8_P	730
	**Glycosphingolipids**		
15	*N*-[eicosanoyl]-1-beta-lactosyl-sphinganine	C_50_H_97_NO_13_	920
16	*N*-[docosanoyl]-eicosasphinganine-1-phospho-[1′-myo-inositol]	C_48_H_96_NO_11_P	894
17	*N*-[octadecanoyl]-1-beta-lactosyl-sphinganine	C_48_H_93_NO_13_	892
18	1-[11*Z*-eicosenoyl]-2-[13*Z*,16*Z*-docosadienoyl]-glycero-3-phosphocholine	C_50_H_94_NO_8_P	868
19	*N*-[docosanoyl]-sphinganine-1-phospho-[1′-myo-inositol].	C_46_H_92_NO_11_P	866
20	*N*-[2-hydroxyheptacosanoyl]-1-*O*-beta-d-glucosyl-15-methylhexadecasphing-4-enine	C_50_H_97_NO_9_	856
21	*N*-hexacosanoyl-1-*O*-beta-D-glucosyl-4-hydroxy-15-methylhexadecasphinganine	C_49_H_97_NO_9_	844
22	*N*-[2-hydroxyhexacosanoyl]-1-*O*-beta-d-glucosyl-15-methylhexadecasphing-4-enine	C_49_H_95_NO_9_	842
23	*N*-[2R-hydroxypentacosanoyl]-1-beta-glucosyl-4E,8*Z*-octadecasphingadienine	C_49_H_93_NO_9_	840
24	*N*-[17*Z*-hexacosenoyl]-1-beta-glucosyl-sphing-4-enine	C_50_H_95_NO_8_	838
25	*N*-pentacosanoyl-1-*O*-beta-d-glucosyl-4-hydroxy-15-methylhexadecasphinganine	C_48_H_95_NO_9_	830
26	1-eicosyl-2-eicosyl-sn-glycero-3-phosphocholine	C_48_H_100_NO_6_P	818
27	*N*-tetracosanoyl-1-*O*-beta-d-glucosyl-4-hydroxy-15-methylhexadecasphinganine	C_47_H_93_NO_9_	816
28	*N*-[2-hydroxytetracosanoyl]-1-*O*-beta-d-glucosyl-15-methylhexadecasphing-4-enine	C_47_H_91_NO_9_	814
29	*N*-[15*Z*-tetracosenoyl]-1-beta-glucosyl-sphinganine	C_48_H_93_NO_8_	812
30	1-[3-hydroxyphytanyl]-2-phytanyl-sn-glycero-3-phosphoethanolamine	C_45_H_94_NO_7_P	792
31	1-octadecyl-2-docosanoyl-sn-glycero-3-phosphoethanolamine	C_45_H_92_NO_7_P	790
32	1-[11*Z*,14*Z*-eicosadienoyl]-2-[6*Z*,9*Z*,12*Z*-octadecatrienoyl]-glycero-3-phosphoethanolamine	C_43_H_76_NO_8_P	766
33	1-[9*Z*,12*Z*,15*Z*-octadecatrienoyl]-2-[8*Z*,11*Z*,14*Z*-eicosatrienoyl]-glycero-3-phosphoethanolamine	C_43_H_74_NO_8_P	764
34	*N*-[2-hydroxyhexacosanoyl]-4R-hydroxysphinganine	C_44_H_89_NO_5_	712
35	*N*-[2-hydroxyheptacosanoyl]-15-methylhexadecasphing-4-enine	C_44_H_87_NO_4_	694
36	*N*-[2-hydroxypentacosanoyl]-15-methylhexadecasphing-4-enine	C_42_H_83_NO_4_	666
37	1,2-Dihexadecyl-sn-glycero-3-phosphoethanolamine	C_37_H_78_NO_6_P	664
38	*N*-[2-hydroxy-hexacosanoyl]-tetradecasphing-4-enine	C_40_H_79_NO_4_	638
39	1,2-ditetradecyl-sn-glycero-3-phospho-*N*, *N*-dimethylethanolamine	C_35_H_74_NO_6_P	636
40	*N*-[2R-hydroxyicosanoyl]-8*Z*-octadecasphingenine	C_38_H_75_NO_4_	610
	**Separate clusters of monoacylglycerophosphoethanolamines and derivatives**		
41	**(a)** *N*-[2-hydroxyhexacosanoyl]-sphinganine	C_44_H_89_NO_4_	696
42	*N*-[2R-hydroxyhexadecanoyl]-4R-hydroxy-20E-hexacosasphingenine	C_42_H_83_NO_5_	682
43	**(b)** *N*-[2R,3-dihydroxyhexacosanoyl]-4R-hydroxy-8E-octadecasphingenine.	C_44_H_87_NO_6_	726
44	N-[2-hydroxyhexacosanoyl]-4R-hydroxysphinganine	C_44_H_89_NO_5_	712
45	**(c)** 1-[11*Z*,14*Z*-eicosadienoyl]-2-[9*Z*-tetradecenoyl]-glycero-3-phospho-[1′-sn-glycerol]	C_40_H_73_O_10_P	745
46	1-[11*Z*,14*Z*-eicosadienoyl]-2-dodecanoyl-glycero-3-phospho-[1′-sn-glycerol]	C_38_H_71_O_10_P	719
47	**(d)** 1-[11*Z*,14*Z*-eicosadienoyl]-glycero-3-phosphoethanolamine	C_25_H_48_NO_7_P	506
48	2-[11*Z*,14*Z*,17*Z*-eicosatrienoyl]-sn-glycero-3-phosphoethanolamine	C_25_H_46_NO_7_P	504

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
