# Peer review of "Comprehensive GCMS and LC-MS/MS Metabolite Profiling of Chlorella vulgaris"

_marinedrugs, 2020, doi:10.3390/md18070367_

Round 1
Reviewer 1 Report
The work is very technical, making it difficult to read, which is a kind of weakness in this article. On the other hand, it describes in a very detailed way the identification of individual compounds, which can undoubtedly be useful for those involved in the analysis of similar metabolites.
In my opinion, the text is not scientific enough - it does not create any research problem - it does not solve the research hypothesis / does not explain the mechanism etc.
The authors point to the potential or already known uses of Chlorella based on identified metabolites - but this is not enough to name a scientific article.
However, it falls within the scope of the Marine drugs magazine so there is no reason to suggest moving to another journal.
Author Response
Response to reviewers
Reviewer 1:
We greatly appreciate all comments by the reviewer. The followings are our point-by-point responses:
- The work is very technical, making it difficult to read, which is a kind of weakness in this article. On the other hand, it describes in a very detailed way the identification of individual compounds, which can undoubtedly be useful for those involved in the analysis of similar metabolites.
Response: Thank you for your positive comments. Yes, we realize that the work involved was indeed very technical. However, we did our best to present the work in an acceptably scientific manner, thus some very technical descriptions could not be avoided.
- In my opinion, the text is not scientific enough - it does not create any research problem - it does not solve the research hypothesis / does not explain the mechanism etc.
Response: With due respect to your comment, we understand the weakness in the present work. However, the work is a first step in a bigger study, and such preliminary works are nowadays characterized with lack of enough substance. Nevertheless, preliminary works are mandatory to setup a good pace for the main objectives to be achieved. With this regard, we decided to make the profiling of Chlorella vulgaris more comprehensive in details so that the information is worthy of publication and more importantly, useful to the general scientific community for future work on the species and other related species.
- The authors point to the potential or already known uses of Chlorella based on identified metabolites - but this is not enough to name a scientific article.
Response: We pointed out the potential and known uses as a means to justify the present work. As mentioned in response to comment 2, the work is the first step towards a bigger study, which is on further identification of immune stimulating metabolites of Chlorella vulgaris, for which having a more comprehensive knowledge on the metabolites present in the species is crucial to the successful outcome of the study. No doubt that the results of the present study lent support to previous identifications but in addition it also uncovered numerous other metabolites not reported previously from the species.
- However, it falls within the scope of the Marine drugs magazine so there is no reason to suggest moving to another journal.
Response: Thank you so much for your kind comment and consideration. We hope the manuscript can be considered for publication in this prestigious journal.
Reviewer 2 Report
General
I recommend an extensive editing of English language and style. For instance, there are issues regarding: Determiner Use (a/an/the/this, etc.), Incorrect Verb Forms, Pronoun Use, Incorrect Noun Number, Faulty Subject-Verb Agreement, Punctuation, Punctuation in Compound/Complex Sentences, Spelling, Misspelled Words, Confused Words, Enhancement, Word Choice, Style, Passive Voice Misuse, Intricate Text, Wordy Sentences, Sentence Structure, Incomplete Sentences
Descriptive paper. Results need to be compared with current literature.
Relevancy of the paper need to be better explained.
Introduction
Introduction is good in terms of content. However, the writing is not clear.
There is some repetition of same information. For instance, the lines 92-94 and lines 105-110.
Results and discussion
Line 113 till line 119 it’s a description of the FAMEs analysis method. Should be moved to materials and methods section.
Line 113 till line 119. Two m/z are mentioned for each fatty acid group. Which m/z was used as qualifier and/or quantifier?
Line 121 remove “ ]”.
A lot of information of section 2.2 (from 2.2.1 till 2.2.6) is description of the analysis methods. Therefore, this text should be moved to the methods section. Methods should only be mentioned in the results and discussion section if they are compared with other methods and their selection in relation to others is explained, which is not the case of the actual manuscript.
Fig.2 It should be added in the figure legend what the numbers above each peak represent or where this information can be found. The two chromatograms are not the same size and the axes titles are blurry and difficult to read. Esthetical changes needed.
In Fig.1 the letter labels (A, B, C D) are in the right side and in Fig. 2 they are in the left. Choose one style and keep it through the manuscript.
Table 2. A description of table 2 columns names should be added to the table 2 legend.
Does m/z [M+H]+ of Table 3 means the same of [M+H]+ of table 4?
Why no tR [min] is shown in Table 3?
In Table 4 and 5 says that bold and underlined m/z values indicate most intense fragments. Why they need to be both bold and underline? Does only bold or only underline mean something different?
Better descriptions of column names in tables and graphs needed.
Also, there is a need to be consistent in terms of formats and abbreviations through the manuscript.
Line 254 to 256, provide references to “…to the literature”.
Line 267 (please check in the manuscript, same error occurs other times) Instead of [44][45][46][47] it should be [44 - 47].
Fig. 6 Low picture quality/blurry.
Fig.6 The color of legend of the green and orange circle do not match the color of the figures.
Table 7. Information should be aligned.
Fig. 8 Fuzzy
Fig. 7, Fig. 9 and Fig. 11 Be consistent with figures structure. In Fig.7 “1.” And in Fig. 9 “(2).”
Can resolution of chromatograms be improved?
Compare the results of Chlorella vulgaris metabolite profile with other microalgae or with Chlorella vulgaris under different culture conditions.
Can quantification and not only detection be performed for some of the metabolites?
Material and Methods
If a standardize knowledge of Chlorella vulgaris metabolome under “normal” culture conditions was the main aim of the study, why did you use a local strain and not a more studied strain that more researchers would have access to?
Why was the strain cultured in duplicates and not triplicates?
Line 353 “The parameter for the algal culture was adopted from the original BBM media with slight modification.” Explain what was the modification.
Section 3.3. Was the rupture of cells walls confirmed by microscopy?
How many technical replicates were used?
Author Response
Response to reviewers
Reviewer 2:
We greatly appreciate all comments by the reviewer. The followings are our point-by-point responses:
General
- I recommend an extensive editing of English language and style.
Response: Thank you for your observation. We did our best to edit the manuscript and make the entire message to be more understood.
Descriptive paper
- Results need to be compared with current literature.
Response: Again, we thank you for your comment. We feel that many comparisons with current literatures have been made in the tables where results were presented. The comparison was also made in results and discussion section. In addition, newly reported metabolites were also mentioned as not previously detected in C.vulgaris. Nevertheless, many additions were made to make the comparisons more obvious (see lines: 133-134; 212-213; 237-238; 252-253; 271; 319).
- Relevancy of the paper need to be better explained.
Response: The relevancy of the paper was further clarified in lines 107-111 of the revised manuscript.
Introduction
- Introduction is good in terms of content. However, the writing is not clear.
Response: Thank you for the positive comment. As per our response to Comment 1, we assure you that we have done our best to edit the manuscript and make the entire message to be more understood.
- There is some repetition of same information. For instance, the lines 92-94 and lines 105-110.
Response: With due respect to your observations the context was not the same. This is because the sentences in lines 92-94 is part of statement of the problem while lines 105-110 is part of justification of the study.
Results and discussion
- Line 113 till line 119 it’s a description of the FAMEs analysis method. Should be moved to materials and methods section.
Response: Moved as suggested to section 3.4 under Material and methods.
- Line 113 till line 119. Two m/z are mentioned for each fatty acid group. Which m/z was used as qualifier and/or quantifier?
Response: Both these m/z signals were used for the identification of fatty acids, but not for quantification. In this study, to calculate the ratio of fatty acids, area obtained from total ion chromatogram was used as mentioned in the materials and method section.
- Line 121 remove “ ]”.
Response: Typographical error: the “]” was removed.
- A lot of information of section 2.2 (from 2.2.1 till 2.2.6) is description of the analysis methods. Therefore, this text should be moved to the methods section. Methods should only be mentioned in the results and discussion section if they are compared with other methods and their selection in relation to others is explained, which is not the case of the actual manuscript.
Response: thank you for pointing this out. We have moved the relevant texts to the appropriate sections under Material and methods.
- Fig.2 It should be added in the Figure legend what the numbers above each peak represent or where this information can be found. The two chromatograms are not the same size and the axes titles are blurry and difficult to read. Esthetical changes needed.
Response: The information has been added to Figure 2 legend. We have also corrected and adjusted the size and quality of the figure.
- In Fig.1 the letter labels (A, B, C D) are in the right side and in Fig. 2 they are in the left. Choose one style and keep it through the manuscript.
Response: Thank you. The labels on the Figures have been adjusted as advised.
- Table 2. A description of table 2 columns names should be added to the table 2 legend.
Response: Thank you. The description has been added as advised.
- Does m/z [M+H]+ of Table 3 means the same of [M+H]+ of table 4?
Response: Yes, they are the same.
- Why no tR [min] is shown in Table 3?
Response: The retention times have been added.
- In Table 4 and 5 says that bold and underlined m/z values indicate most intense fragments. Why they need to be both bold and underline? Does only bold or only underline mean something different?
Response: The most intense fragments are now just underlined.
- Better descriptions of column names in tables and graphs needed.
- Also, there is a need to be consistent in terms of formats and abbreviations through the manuscript.
Response: Thank you. We have made improvements to all these aspects as advised and hope that they are now more satisfactory.
- Line 254 to 256, provide references to “…to the literature”.
Response: The statement meant that the supposedly identified adducts were rejected because there are large error values resulting from comparison of exact m/z values (from online databases) and m/z values from our results. It is a common practice to reject error values greater than m/z 0.2.
- Line 267 (please check in the manuscript, same error occurs other times) Instead of [44][45][46][47] it should be [44 - 47].
Response: The format has been checked and corrected throughout the manuscript.
- Fig. 6 Low picture quality/blurry.
- Fig.6 The color of legend of the green and orange circle do not match the color of the figures.
Response: Response: We have tried to improve the quality by increasing the image to 800 pixels. The exact colours on the figures are not available on power point. Nevertheless, we have simplified the figure and improved presentation so that they can clearly illustrate the whole network generated from the MS data. We hope that it is now more satisfactory.
- Table 7. Information should be aligned.
Response: The information has been aligned.
- Fig. 8 Fuzzy
Response: Figure 8 has been reformatted to look neater.
- Fig. 7, Fig. 9 and Fig. 11 Be consistent with figures structure. In Fig.7 “1.” And in Fig. 9 “(2).” Can resolution of chromatograms be improved?
Response: We have corrected the inconsistencies. However, we are not able to improve further on the resolution of the chromatograms.
- Compare the results of Chlorella vulgaris metabolite profile with other microalgae or with Chlorella vulgaris under different culture conditions.
Response: We thank you for the comment. However, it is believed that there must be differences between microalgae cultured under different media formulations. Nevertheless, observable differences were discussed in presenting results of our findings, especially with respect to the carotenoids and other pigments which were not present in other strains (references provided). We felt that it is not necessary to include a separate heading on that issue as substantial discussions of results addressed it. In addition, we addressed this issue in detail about Chlorella vulgaris metabolite profile as a subject of our other prepared publication.
- Can quantification and not only detection be performed for some of the metabolites?
Response: Yes, quantification could be possible, subject to availability of standards. However, due to constraints in obtaining pure standards, we have not included it among our objectives of the present study, which is specific on qualitative profiling. Incidentally, the relative quantification of some of the bioactive metabolites has been carried out and is the subject of another publication which is being prepared.
Material and Methods
- If a standardize knowledge of Chlorella vulgarismetabolome under “normal” culture conditions was the main aim of the study, why did you use a local strain and not a more studied strain that more researchers would have access to?
Response: We also thank you and appreciate your comment here. The issue is of paramount concern. However, the research was sponsored under a grant from JICA through Ministry of Education Malaysia. The aim of the said project is to improve aquaculture in Malaysia using microalgae as immunostimulants which necessitate us to culture and use a local strain for the entire research.
- Why was the strain cultured in duplicates and not triplicates?
Response: The strain was cultured in duplicates due to constraints in getting access to adequate units of bioreactors at the time of the experiment. The facility was shared by a very big number of researchers.
- Line 353 “The parameter for the algal culture was adopted from the original BBM media with slight modification.” Explain what was the modification?
Response: The modification was in the temperature used, which we could not control or maintain at 28 oC. The modification was added to the sentence (see Line 365 – 366 of the revised manuscript).
- Section 3.3. Was the rupture of cells walls confirmed by microscopy?
Response: No, we did not think of it at that moment. Nevertheless, we assume that the cell walls were ruptured as a result of solvent extraction considering the detected metabolites obtained from our findings would not have been accessed if otherwise. But we thank you for this suggestion and we will apply such confirmation in our future researches.
- How many technical replicates were used?
Response: Thank you for your comment. However, we felt that no technical replicates were necessary in culturing microalgae for mass spectrometry. Nevertheless, two or three replicates are recommended in case of microbial contamination, but not compulsory.